# Aggressiveness of *Puccinia striiformis* f. sp. *tritici* Isolates at High Temperatures: A Study Case in Core Oversummering Area of Gansu as Inoculum Source

**DOI:** 10.3390/plants13243518

**Published:** 2024-12-16

**Authors:** Bo Zhang, Jie Zhao, Jin Huang, Xiaojie Wang, Zhijie Guo, Qiuzhen Jia, Shiqin Cao, Zhenyu Sun, Huisheng Luo, Zhensheng Kang, Shelin Jin

**Affiliations:** 1State Key Laboratory for Crop Stress Resistance and High-Efficiency Production, College of Plant Protection, Northwest A&F University, Yangling 712100, China; zhangbo@gsagr.cn (B.Z.); jiezhao@nwsuaf.edu.cn (J.Z.); wangxiaojie@nwsuaf.edu.cn (X.W.); 2Plant Protection Institute, Gansu Academy of Agricultural Sciences, Lanzhou 730070, China; huangjin@gsagr.cn (J.H.); guozhijie@gsagr.cn (Z.G.); jiaqiuzhen@gsagr.cn (Q.J.); caoshiqin@gsagr.cn (S.C.); sunzhy@gsagr.cn (Z.S.); luohsh@gsagr.cn (H.L.)

**Keywords:** wheat stripe rust, aggressiveness, high temperature, disease severity, spore germination, latent period, oversummering region

## Abstract

Wheat stripe rust, caused by a biotrophic, obligate fungus *Puccinia striiformis* f. sp. *tritici* (*Pst*), is a destructive wheat fungal disease that exists worldwide and caused huge yield reductions during pandemic years. Low temperatures favor the development of the disease, but the global average temperature has been increasing since 1850, especially in China, which has a higher rising rate than the global average. In the last two decades, *Pst* isolates have shown increased aggressiveness under high temperatures. However, the effect of rising temperatures on the aggressiveness of *Pst* has remained unknown in China. Therefore, this study assessed the aggressiveness of 15 representative *Pst* isolates (6 new isolates collected before 2016 and 9 old isolates collected after 2016) in Gansu under high temperatures by measuring and comparing disease severity, spore germination, and latent period on wheat seedlings at 16 °C, 18 °C, and 22 °C. The results indicated that **the** six new isolates showed greater disease severity, higher spore germination ratio, and shorter latent period than the nine old isolates, indicating that the new isolates were more aggressive under high temperatures than the old isolates. Some new isolates, such as CYR34, CYR33, and CYR32, which are predominant, were inferred to be associated with high-temperature adaptation in addition to having more susceptible hosts. Our results provided an insight into changes in *Pst* isolates at warmer temperatures and increasing incidence of wheat stripe rust in China, especially in eastern sporadic epidemiological areas in recent years. Thus, the new isolates are likely to be a potential risk for causing increasing stripe rust incidence.

## 1. Introduction

Wheat stripe rust, caused by *Puccinia striiformis* f. sp. *tritici* (*Pst*), is a destructive wheat disease worldwide. Outbreaks of the disease can lead to significant yield reductions [1,2]. In China, it has caused eight severe nationwide epidemics, resulting in total yield losses of 13.89 million metric tons since the 1950s [3]. Compared with many plant pathogenic fungi, low temperatures favor the development of the rust fungus. Under optimum conditions, the rust starts to sporulate 12–13 days after infection [4]. However, high temperatures can inhibit sporulation and force *Pst* to become dormant. Host plants are susceptible at the favorable temperature of 13 °C but showed resistance at 25 °C [5]. Due to the biotrophic parasitic nature of *Pst*, oversummering is crucial for the survival of *Pst* during summer to succeed in its disease cycle.

In China, in recent years, ~24 million hectares of wheat are grown annually, and an annual average of 1.33 million hectares of wheat is infected by stripe rust [6]. Gansu is the core oversummering region for *Pst* and the key inoculum source region of wheat stripe rust. In this region, the annual average wheat-growing area was 0.73 million hectares, and 0.10–0.35 million hectares of wheat have been infected by stripe rust in recent years [6]. Historically, most new races (or variants) first emerged in this region nationwide, and the breakdown of main cultivated wheat cultivars was first observed in this region [3]. Importantly, it serves as the source of inoculum towards wheat-growing regions of Huang-Huai-Hai and eastern China in autumn and spring, whereas the rust is unable to survive in these regions in summer due to the limitation of high temperatures [7]. Therefore, inoculum in Gansu is critical for influencing the occurrence of disease all year round in main wheat production regions in the whole country [8,9,10].

Based on annual surveillance of *Pst* races in the past decades [11,12,13,14], CYR32, CYR33, and CYR34 are predominant races among *Pst* populations in Gansu. Among these, CYR33 and CYR32 were predominant before 2014, and since then, there has been a minor decrease in occurrence frequency. CYR34 (previously G22-9, with virulence to differential line Guinong 22) has combined virulence to *Yr26* and *Yr10* [15]. It was first identified in Gansu in 2010, and by 2012, it had become the third most predominant race, next to CYR33 and CYR32. Since 2014, it has asserted its dominance not only in the region but also in the entirety of China [16]. Su11 (virulent to differential line Suwon 11) and G22 (virulent to differential line Guinong 22) were significant additions to the three predominant races. Su11-4, Su11-5, and Su11-7 were the third or fourth most predominant after CYR33 and CYR32 from 2004–2008 and have been detected annually since 2008, with a low occurrence frequency. G22-14 was identified in 2011 and constantly had a low occurrence frequency, ranging from 7.40% in 2011 to 6.0% in 2020 [16]. Recently, Zhou et al. [17] identified that CYR34, G22-14, CYR32, and CYR33 were the top four predominant races among Chinese *Pst* populations from 16 provinces in 2021; each of these races were detected in the most provinces of China [17].

Wheat stripe rust has been becoming increasingly severe in certain sporadic epidemiological regions or warmer regions in some countries worldwide since 2000. The rust can spread for a long distance from its origin by air currents and colonize to form multiple populations in the local area [2]. In the last two decades, a global dispersion of *Pst* isolates has been reported in eastern United States, Australia, and other regions, as well as increased disease severity in these regions [18]. In the United States, the disease is generally prevalent in the Pacific Northwest. However, since 2000, it has become a significant issue in the south–central states [19]. Subsequently, Milus et al. [20] reported that, in the eastern and south–central United States, the new isolates collected before 2000 were more aggressive under high temperatures than the old isolates collected after 2000 at 18 °C but not at 12 °C. These isolates were responsible for recent disease epidemics in these states [18,19,20]. In Australia, Loladze et al. [21] suggested that *Pst* pathotypes detected after 2002 were not significantly affected by contrasting temperatures in incubation (10 °C and 15 °C) and latent periods (17 °C and 23 °C). This indicates they were not responsible for the wheat stripe rust epidemic in 2002, but other factors might have an impact on disease epidemics [21]. De Vallavieille-Pope stated that *Pst* isolates PstS1 and PstS2 likely originated from East Africa and subsequently spread to many other countries worldwide. They displayed stepwise adaptation to high temperatures as they migrated from cool to warm regions of the continents [22,23]. Likewise, in Europe, the “Warrior” isolates spread widely in warm regions. It is suggested that they might adapt to warm temperatures in that they are effective in causing infection and reducing the latency period when moving between cool and warm temperatures. The change in these isolates’ relationship to temperatures poses a potential threat of severe epidemics of wheat stripe rust under warming climate conditions [22]. In recent years (2017, 2019, and 2020), severe wheat stripe rust has occurred in the main wheat-growing regions of China, especially in sporadic epidemiological regions in the east [3,24,25], such as Zhejiang, in which the disease is regularly inhibited at the end of March due to warming temperature. However, it had an extended duration until the end of April [25]. More importantly, we demonstrated that invasive inoculum in the eastern region mainly migrated from the Gansu oversummering region. Therefore, we hypothesized that the incidents of the stripe rust disease in China could be attributed to *Pst* isolates that have adapted to high temperatures, as those reported in the United States and elsewhere, demonstrating increased aggressiveness under warm conditions. Thus, we aimed to determine whether the new isolates of the rust are more aggressive than old ones in high temperatures in the Gansu oversummering region as inoculum origin.

## 2. Results

Based on pre-experimental tests, 50 *Pst* isolates from 2005 to 2019 showed greater development of stripe rust infection under high temperatures than regular temperatures, fifteen of which, with consistent repeatability, were used as representative isolates for evaluation in high temperatures in the present study based on occurrence frequency during different years in different locations (Table 1).

### 2.1. Disease Severity

Under incubation at 16 °C, infected 10-day-old “Mingxian 169” wheat seedlings for 16 days post-inoculation (dpi) showed that disease severity (DS) of overall isolates on leaves of wheat seedlings ranged from 30.3% to 48.5%, with an average of 40.6% (average value of data of disease severity in three repetitions) (Table 2; Figure 1). The isolate Su11-5 (GS-3) collected in 2005 had the minimum DS (30.3%). However, the isolate CYR34 collected in 2016 (GS-12) exhibited the maximum DS (48.5%). The DS of seven isolates, including CYR33 (GS-4, and -8), Su11-4 (GS-6), Su11-35 (GS-7), CYR32 (GS-9 and -11), and CYR34 (GS-12), exceeded 40% (41.7–48.5%) and was higher than those of the other eight isolates and the average value of all isolates (40.6%). Four isolates collected in 2005, Su11-4 (GS-1), Su11-7 (GS-2), Su11-5 (GS-3), and CYR32 (GS-5), showed a lower DS (30.3–39.5%) than the average value. Most of isolates collected after 2010, except for CYR33 (GS-10) and CYR34 (GS-14 and -15), had a higher DS than the average value.

A significant difference was observed among seven isolates, including Su11-5 (GS-3), Su11-35 (GS-7), CRY33 (GS-8), CYR32 (GS-11), CYR34 (GS-12 and -14), and G22-14 (GS-13) (*p* < 0.05), but this was not found among isolates Su11-4 (GS-1 and -6), Su11-7 (GS-2), CYR33 (GS-4 and -10), CYR32 (GS-5 and -9), and CYR34 (GS-15) and among Su11-35 (GS-7), CYR32 (GS-11), CYR34 (GS-12), and G22-14 (GS-13) (*p* < 0.05) (Table 2).

At 22 °C, the DSs of isolates developing on leaves of wheat seedlings at 16 dpi were measured and indicated that, overall, isolates had a DS range of 9.5-35.0%, with an average of 22.1% (Table 1). Isolates Su11-5 (GS-3) showed the minimum (9.5%) DS on wheat seedlings while CYR34 (GS-12) showed the maximum (35.0%). Nine isolates, including CYR33 (GS- 4, -8, and -10), CYR32 (GS-5, -9, and -11), CYR34 (GS-12 and -15), and G22-14 (GS-13), displayed a higher DS than the average value (22.1%), with a range of 23.0% to 35.0%, whereas the remaining six isolates Su11-4 (GS-1 and -6), Su11-7 (GS-2), Su11-5 (GS-3), Su11-35 (GS-7), and CYR34 (GS-14) had a DS range of 9.5% to 17.7%, lower than the average value.

A significant difference was observed among isolates (*p* < 0.05). However, there was no significant difference among isolates Su11-4 (GS-1 and -6) and Su11-35 (GS-7), between Su11-7 (GS-2) and Su11-5 (GS-3), among CYR32 (GS-4, -5, and -9), CYR33 (GS-10), and CYR34 (GS-15), between Su11-4 (GS-6) and Su11-35 (GS-7), between CYR32 (GS-11) and CYR34 (GS-12), and between CYR33 (GS-8) and G22-14 (GS-13) (*p* < 0.05).

Sixteen days after inoculation with urediniospores and incubation at 16 °C, the isolates collected before 2010 (old isolates) displayed variable DS size on infected wheat seedling leaves. CYR33 (GS-12) exhibited the largest DS, followed by CYR32 (GS-5). Su11-5 (GS-3) showed the smallest DS (Table 2). Among the isolates collected after 2010 (new isolates), CYR34 (GS-12) showed the largest DS, followed by G22-14 (GS-13). Meanwhile, CYR34 (GS-14) had the smallest DS.

Regardless of incubation at 16 °C or 22 °C, the most isolates post-2010 showed a DS higher than those pre-2010 (Table 2; Figure 1), indicating that the new isolates could have evident adaptation to high temperatures compared to the old isolates.

Based on the collection year, the DSs of the isolates collected in 2005, 2010, and post-2016 on leaves of wheat seedlings at 16 °C were 37.3%, 42.3%, and 42.3%, and those at 22 °C were 17.7%, 21.1%, and 27.5%, respectively (Table 3; Figure 1). Analysis of variance (ANOVA) showed that there was a significant difference in DS among these three subgroups at both 16 °C and 22 °C (*p* < 0.05) (Table 3; Figure 1). At 16 °C, a significant difference was observed in 2005 and 2010 subgroups and 2005 and post-2016 subgroups, but it was not observed between 2010 and post-2016 subgroups (*p* < 0.05). At 22 °C, a significant difference was noted between 2005 and post-2016 subgroups and post-2016 and 2010 subgroups, but it was not observed between 2005 and 2010 subgroups (*p* < 0.05).

Analysis of variance (ANOVA) indicated that there was a highly significant difference between and within both temperature isolates (16 °C and 22 °C), and among overall isolates (*p* < 0.001). The interactions between temperature × year and temperature × isolate were also significant (*p* < 0.05) (Table 4; Figure 1).

### 2.2. Spore Germination

The germination percentage of urediospores (GPU) of all isolates at 4, 8, 12, and 24 h at 16 °C had a range of 0–13.7%, 0–18.7%, 0–31.7%, and 0.1–59.6%, respectively (Table 5). The result indicated that, the longer incubation duration, the higher the GPU was. Among all isolates, CYR34 (GS-12) exhibited the maximum GPU at 4, 12, and 24 h, but not 8 h, at 16 °C. With the extension of the incubation duration, except for Su11-7 (GS-2) and Su11-5 (GS-3), all isolates exhibited gradually increased GPU and reached the peak at 24 h. At 24 h, eight isolates including Su11-4 (GS-1 and -6), Su11-7 (GS-2), Su11-5 (GS-5), Su11-35 (GS-7), CYR33 (GS-10), CYR32 (GS-11), and CYR34 (GS-15) had a GPU lower than the average value (26.7%). However, GPU of the other seven isolates, CYR33 (GS-4 and -8), CYR32 (GS-5 and -9), G22-14 (GS-13), and CYR34 (GS-12 and -14), was greatly higher than the average value (Table 5; Figure 2).

At 24 h of incubation at 16 °C, among the isolates collected in 2005, the maximum GPU was observed in CYR32 (GS-5), followed by CYR33 (GS-4), and the minimum GPU in Su11-5 (GS-3) (Table 5). Among the isolates collected in 2010, CYR33 (GS-8) had the highest GPU, followed by CYR32 (GS-9), and Su11-4 (GS6) had the lowest. Among the isolates collected post-2016, the GPU of CYR34 (GS-14) was the maximum, followed by G22-14 (GS-13), and the minimum was that of CYR34 (GS-15). ANOVA indicated that most of the isolates, except for Su11-4 (GS-1), Su11-5 (GS-3), Su11-7 (GS-2), Su11-4 (GS-6), and Su11-35 (GS-7), were significantly different in GPU at 24 h (*p* ≤ 0.05). However, no significant difference was observed among some isolates at each of the four time points (Table 5; Figure 2).

The same races collected in different years showed different GPUs for 24 h at 16 °C (Table 5; Figure 2). For example, CYR33 (GS-4, -8, and -10) had GPUs of 32.9%, 43.9%, and 24.6%, respectively. Likewise, CYR 34 (GS-12, -14, and -15) showed GPUs of 59.6%, 46.2%, and 15.3%, respectively.

Based on the collection year of isolates, the GPU of the isolates collected in 2005 was 20.7%, for those in 2010 it was 21.3%, and for those after 2016 it was 38.0% (Table 6), showing variable GPU among the three subgroups. ANOVA displayed that there was a significant difference among subgroups (*p* < 0.05). A significant difference was observed between post-2016 and 2005 subgroups and between post-2016 and 2010 subgroups, but not between 2005 and 2010 subgroups (*p* < 0.05) (Table 6; Figure 2).

### 2.3. Latent Period

After inoculation with urediospores, the latent period of all isolates on leaves of susceptible wheat cv. “Mingxian 169” seedlings ranged from 9.7 d to 11.3 d after incubation for 16 d at 18 °C and from 10.2 d to 12.2 d after incubation for 16 d at 22 °C (Table 7). At 18 °C, nine isolates, CYR33 (GS-4, -8, and -10), CYR32 (GS-5, -9, and -11), CYR34 (GS-14 and -15), and G22-14 (GS-13), showed a latent period shorter than or equal to the average value (10.5 d). However, the remaining six isolates, Su11-4 (GS-1 and -6), Su11-7 (GS-2), Su11-5 (GS-3), Su11-35 (GS-7), and CYR34 (GS-12), had a longer latent period than the average value.

At 22 °C, nine isolates, CYR33 (GS-4, -8, and -10), CYR32 (GS-5, -9, and -11), CYR34 (GS-14 and -15), and G22-14 (GS-13), had a latent period shorter than the average value (11.1 d), indicating they were more adaptable to high temperatures, whereas the remaining six isolates, including Su11-4 (GS-1 and -6), Su11-7 (GS-2), Su11-5 (GS-3), Su11-35 (GS-7), and CYR34 (GS-12), exhibited a longer latent period than the average value (Table 7; Figure 3).

A significant difference was observed among all isolates except Su11-4 (GS-1), Su11-5 (GS-3), Su11-4 (GS-6), CYR33 (GS-4, -8, and -10), CYR32 (GS-5), and CYR34 (GS-13) and between Su11-35 (GS-7) and CYR34 (GS-12) at 18 °C (*p* ≤ 0.05) (Table 8). Likewise, there was a significant difference among most of the isolates after incubation for 16 d at 22 °C (*p* ≤ 0.05). However, no significant difference was observed among Su11-4 (GS-1 and -6), Su11-5 (GS-3), Su11-35 (GS-7), and CYR34 (GS-12) and between CYR33 (GS-4, -8, -10, and -14), CYR32 (GS-5 and -11), G22-14 (GS-13), and CYR34 (GS-15) (*p* ≤ 0.05) (Table 7; Figure 3).

Among isolates from 2005, the minimum latent period at both 18 °C and 22 °C was observed in CYR33 (GS-4). In isolates from 2010, CYR32 (GS-9) exhibited the shortest latent period at both temperatures. In the post-2016 isolate group, the minimum latent period was shown by CYR 34 (GS-14 and -15) at 18 °C and CYR32 (GS-11) at 22 °C (Table 7; Figure 3).

Based on the collection year of isolates, the overall latent period of the isolates collected in 2005 (2005 subgroup) at 18 °C was 10.38 d, for those in 2010 (2010 subgroup) it was 10.65 d, and for those post-2016 (post-2016 subgroup) it was 10.37 d (Table 8). At 18 °C, the latent period of the isolates collected in 2010 was longer than for those in 2005 (2005 subgroup), but shorter than for those from post-2016 (post-2016 subgroup). At 22 °C, the latent period of 2005 and post-2016 subgroups was 11.00 d, longer than those in the 2010 subgroup. ANOVA displayed that there was no difference in the latent period among subgroups from different years at 18 °C and 22 °C (*p* < 0.05) (Table 8; Figure 3).

## 3. Discussion

This study testified that new isolates (after 2010) of *Pst* were more aggressive than old isolates (before 2010) in the Gansu oversummering region in China. CYR34, CYR32, CYR33, and G22-14 rapidly developed to be dominant in the Chinese *Pst* population since their initial emergence. Now, they are the most predominant race groups nationwide [15,16,27,28]. Su11-4, Su11-5, and Su11-7, with virulence to wheat differential line Suwon 11, have been minor race groups with a low frequency in the *Pst* population since 2004 in Gansu and other wheat-growing areas of China [11,12,13,14,16]. The results of the present study indicated that new isolates showed a higher average disease severity at 16 °C and 22 °C, a higher average germination percentage of urediospores at 16 °C, and shorter length of latent period at 18°C and 22 °C (except for GS-12) than old isolates. Some molecular studies showed high genetic variation in the Gansu *Pst* population [29,30], and genetic mutation may have occurred between isolates that may have enhanced isolate fitness against environmental factors such as temperature. However, further studies need to be conducted to determine whether this adaptation to high temperature occurred due to genetic variation within *Pst* population or due to host resistance genes that became ineffective due to other environmental changes. Similar results were previously reported by Milus et al. [17,18] that USA *Pst* races collected after 2000 were more aggressive and better adapted to warmer temperatures than those collected before 2000, which were responsible for the increased severity of stripe rust epidemics that occurred in south–central and eastern regions in recent years [19,20]. Also, similar findings reported by Hovmøller et al. [31] showed the adaptation of *Pst* isolates to high temperature in Central and Northern Europe from 2000–2014 and subsequently revealed that the new isolates have replaced the old isolates from Northwest European populations post-2011, which could be responsible for stripe rust incidences in recent years [31]. A recent study (Awais et al. 2024, under review) also showed that temperature played an important role in shaping the population structure of *Puccinia striiformis*; in that study, a comparative study of *Pst* was performed by considering its interaction with different temperature zones and geographical distance (Xinjiang and Central Asia regions). The result showed that regions with the same temperature had the same population and gene flow, while the regions with temperature differences had population differences and between them there was genetic divergence. In China, in recent years, frequent epidemics of stripe rust have occurred in eastern regions [7,25]. CYR34, CYR33, and Su11 virulent races were detected in the eastern region that migrated from the Gansu oversummering area, which was the main reason for the epidemic in 2019 in this region [7].

CYR34, CYR32, CYR33, and G22-14 races, which were predominant races in recent years in Gansu and other wheat-planting regions of China [13,14,16], may possess more virulence genes than Su11-4, Su11-5, and Su11-7 and those additional virulence genes may be linked to genes for high aggressiveness.

The survival of oversummering inoculum is associated with wheat stripe rust epidemics and mostly occurs under low temperatures (≤22 °C of 10-day average temperature). Based on 678 meteorological data from 2002–2013, average temperatures in oversummering regions have increased 1.34 °C (1960–2001), resulting in a notable increase in the oversummering regions [32]. Possibly, the increased temperatures could gradually influence the adaptation of *Pst* isolates to the surrounding environmental conditions. The wide spread of these isolates could be responsible for severe stripe rust in eastern China in recent years.

Increased incidences of wheat stripe rust have taken place in China in recent years. From 2019 to 2024 cropping seasons, the annual affected wheat area was estimated to be up to 2.78 million hectares nationwide. A recent study by Ju et al. [7] demonstrated that genotype lineage analysis indicated inoculum spreading to the eastern parts was mainly originated from the northwest oversummering region, including Gansu [7,33]. This result was in accordance with genomic analysis from Li et al. [34]. On the other side, CYR34 races and its variants were testified to have relatively higher parasitic fitness than currently predominant races CYR32 and CYR33 [35]. Most recently, a study by Hu et al. reported that, in the Gansu oversummering region, spore density of *Pst* isolates increased with average air temperatures ranging from 10 °C to 21 °C and relative humidity from 60% to 85% from May to June [36]. Whether this finding hints that these *Pst* isolates have acquired adaptability to warmer temperatures in this region needs to be studied.

Plant diseases are mostly affected by multiple factors including pathogens, host, and environment. These factors involve parasitic fitness and virulence (pathogenicity), abiotic interactions of pathogens, susceptibility of hosts, and surrounding environmental conditions. Climate warming can remarkably influence biological aspects in relation to population dynamics of pathogens, such as oversummering, overwintering, survival, and population growth rates [37]. It is also likely to impact severely on the status of agricultural crop disease because warmer temperatures influence the host–pathogen interactions, spatiotemporal distribution of plant diseases, geographic distribution of host pathogens, as well as the incidence of plant diseases [1,38].

The change in temperature could be used to explain *Pst* evolution to adapt to warmer temperatures. With the continuous changes in world climate, strengthening the detection of *Pst* isolates with better adaptation to warmer temperatures for in-time management of wheat stripe rust is necessary. This study indicates that the *Pst* races in the Gansu oversummering region had better adaptation to high temperatures, and thus we further hypothesized that high temperatures could be one of the factors affecting the adaptation of *Pst* races. Further experimental tests are needed.

In conclusion, *Pst* isolates collected after 2016 showed more aggression at high temperatures than those collected before 2016, providing an insight into understanding adaptation of the stripe rust to global warming and epidemics and integrated management of wheat stripe rust.

## 4. Materials and Methods

### 4.1. Pst Isolates and Plants

All isolates were stored at the Institute of Plant Protection, Gansu Academy of Agricultural Sciences, Lanzhou and represented the predominant *Pst* races collected from the Gansu region in the past fifteen years from 2005–2019. CYR34 (previously G22-9) isolates of *Pst*, virulent to *Yr26* and *Yr10,* have rapidly developed into a prevalent race group with highly pathogenicity in Gansu, China since 2016 due to rapid development and accumulation, resulting in more severe stripe rust epidemics [11,14]. These isolates collected before and after 2016 were indicated as “old” and “new” races for the comparative studies performed.

Twenty seeds of the wheat cultivar “Mingxiang 169”, highly susceptible to all known Chinese races of *Pst* and without genetic resistance genes, were grown in a plastic pot (8 cm in diameter) filled with potting mix (Inner Mongolia Mengfei Bio-tech Co., Ltd., Hohhot, Inner Mongolia, China) and cultivated for ten days in a growth chamber (RXZ-1000C, Ningbo Jiangnan Instrument Factory, Zhenjiang, China) in a condition-controlled greenhouse.

### 4.2. Inoculation

Spore suspension was made by adding 2 mg urediospores to 2-mL deionized water with 0.05% (vol./vol.) Tween 20 in a 5 mL sprayer. After mixing, it was sprayed on leaves of 10-day-old wheat seedlings of “Mingxian 169” for inoculation. After spraying, the wheat plants were kept for 24 h at 10 °C in the dark in a dew chamber (LT-36VL, Percival, IA, USA). After incubation, the inoculated seedlings were transferred into a growth chamber to cultivate at 13–16 °C under a dual photoperiod regime of 16 h light and 8 h dark until sporulation. Fresh urediospores were collected using a glass collector by a vacuum pump system and stored in a desiccator at 4 °C in a refrigerator until use.

### 4.3. Assessment of Disease Severity

For comparative studies of isolates’ response to high temperatures, 10-day-old seedlings of “Mingxian 169” wheat were inoculated by spraying urediospore suspension at ~30 cm over the plants. The urediospore suspension was made by the method mentioned above. Inoculated plants were placed in the dew chamber for 24 h at 10 °C in the dark. After incubation, plants were transferred to condition-controlled growth chambers for cultivation for 16 days at 16 °C or 22 °C under a diurnal cycle of 16 h light (10,000–15,000 μmol·m^2^·s^−1^ of light intensity) and 8 h dark until the infection developed fully. Disease severity (% of diseased leaf area) was recorded by the six-rating scale described by Kiani et al. [26] 18–20 days after inoculation from an infected leaf with maximum sporulation. Ten infected leaves were measured in each treatment, and three repetitive treatments were used.

### 4.4. Assessment of Spore Germination

All isolates were recovered on highly susceptible wheat cv. “Mingxian 169” and multiplied abundant spores for use. Approximately 2 mg of fresh urediospores were dispersed onto deionized water in a Petri dish using a brush. The dishes were then incubated at 16°C to induce spore germination. At various time intervals (4, 8, 12, and 24 h), the urediospores under investigation were delicately collected using a brush with a flat and round tip measuring 0.5 cm in diameter. These collected spores were then released into water, forming a few droplets on a clean glass slide. Urediospore germination was assessed at each time interval by examining three random microscopic observation fields of view, with at least 50 spores (n ≥ 50) observed per observation field of view, using a light microscope (Olympus, 1X71, Olympus Corporation, Tokyo, Japan). Germination was considered to have occurred when the germ tube extended to at least half the diameter of the urediospore [39]. The average of the data was used as the spore germination rate.

### 4.5. Assessment of Latent Period

Spore suspension was made by adding 5 mg of fresh urediospores to 10 mL deionized water, mixed well, and sprayed on leaves of 10-day-old wheat seedlings in 3 pots (10 plants per pot). Inoculated plants were placed in a dew chamber for 24 h at 10 °C in the dark. After incubation, plants were transferred to two individual growth chambers at 18 °C or 22 °C under conditions of 10,000–15,000 μmol·m^2^·s^−1^ of light intensity and a diurnal cycle of 16 h light and 8 h dark. The latent period was assessed based on time (number of days) from inoculation to the appearance of uredia on half of the seedling plants [20]. Three repetitions were conducted in all experiments.

### 4.6. Statistical Analysis

Experimental data were subjected to analysis of variance (ANOVA) using the MIXED procedure of the R programming language (ver. 3.6.2). The effect of a linear model was established from the fixed effects of temperature and isolate and the random effects of repetition at different times. The data of disease severity, percentage of urediospore germination, and length of latent period were used for ANOVA. In all experiments, the model was constructed for individual isolates.

Pairwise comparisons were used for comparisons between isolates, between years, between isolate and temperature, and between isolate and year using LSMEANS and ESTIMATE statements [19]. *p* values were used for multiple testing correction based on the Tukey–Kramer test [40].

## Figures and Tables

**Figure 1 plants-13-03518-f001:**
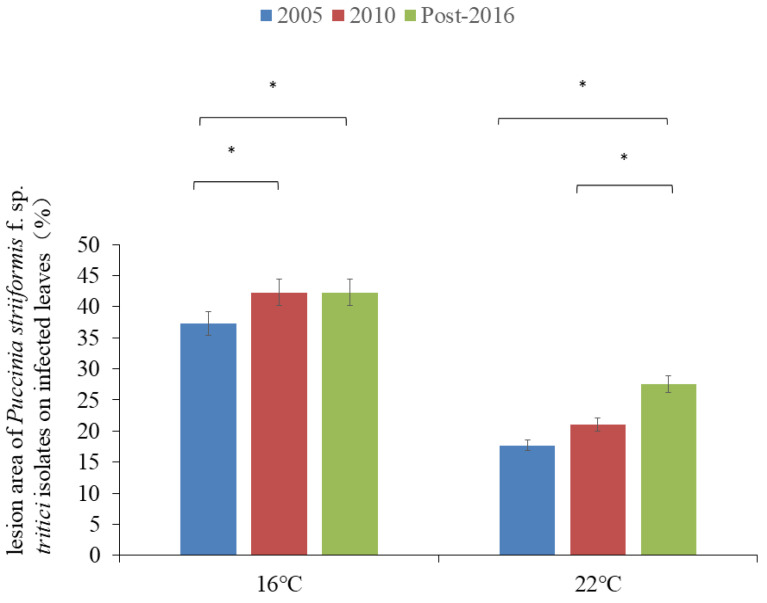
A diagram showing disease severity of *Puccinia striiformis* f. sp. *tritici* isolates collected in 2005, 2010, and post-2016 on infected leaves (%) and significant difference (* *p* < 0.05) at 15 °C and 22 °C, respectively.

**Figure 2 plants-13-03518-f002:**
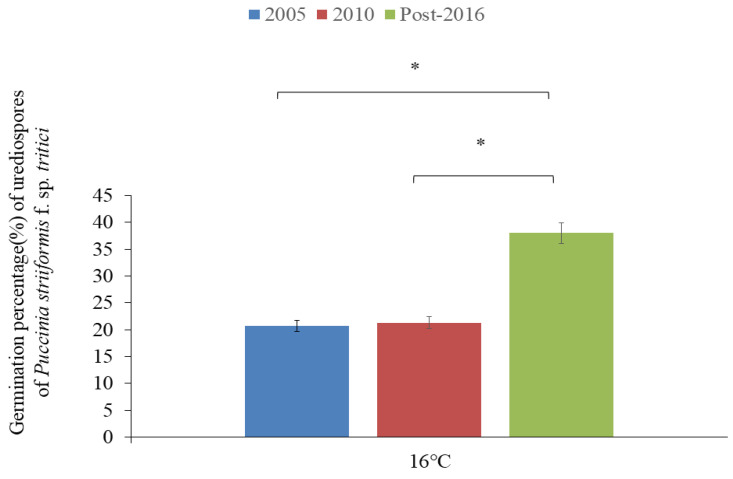
A diagram showing germination percentage (%) of urediospores of *Puccinia striiformis* f. sp. *tritici* isolates collected in 2005, 2010, and post-2016 and significant differences (*p* < 0.05) at 16 °C (* *p* < 0.05).

**Figure 3 plants-13-03518-f003:**
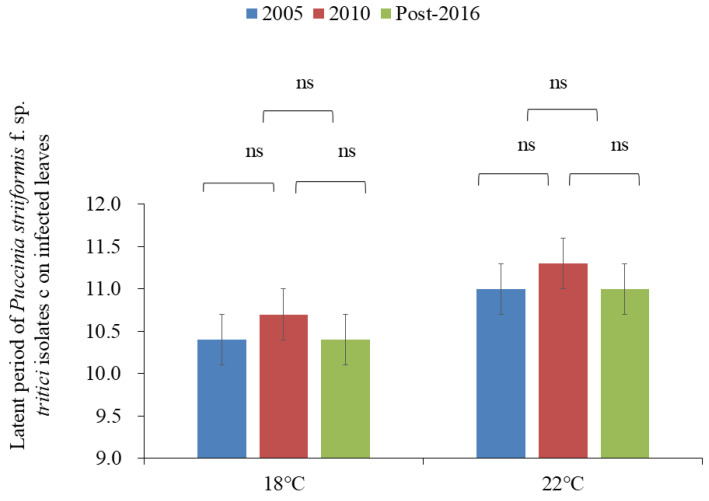
A diagram showing Latent period of *Puccinia striiformis* f. sp. *tritici* isolates collected in 2005, 2010, and post-2016 on infected leaves at 18 °C and 22 °C, respectively. There was no difference (ns) among three subgroups at the two temperatures (*p* < 0.05).

**Table 1 plants-13-03518-t001:** Information of *Puccinia striiformis* f. sp. *tritici* isolates collected in Gansu from 2005–2019.

Code	Isolate ^a^	Location	Occurrence Frequency (%) and Rank in Sampling Year	Altitude (m)	Year
GS-1	Su11-4	Qingshui, Tianshui	5.3 (5)	1700	2005
GS-2	Su11-7	Qinzhou, Tianshui	10.0 (4)	1600	2005
GS-3	Su11-5	Gangu, Tianshui	12.3 (3)	1600	2005
GS-4	CYR33	Lintao, Dingxi	24.1 (1)	1800	2005
GS-5	CYR32	Qinzhou, Tianshui	24.1 (1)	1600	2005
GS-6	Su11-4	Qinzhou, Tianshui	16.08 (3)	1600	2010
GS-7	Su11-35	Qinzhou, Tianshui	9.69 (4)	1862	2010
GS-8	CYR33	Minxian, Dingxi	19.06 (2)	1900	2010
GS-9	CYR32	Zhangxian, Dingxi	19.69 (1)	1900	2010
GS-10	CYR33	Gangu, Tianshui	5.63 (4)	1700	2016
GS-11	CYR32	Qingshui, Tianshui	9.12 (2)	1400	2016
GS-12	CYR34	Qingshui, Tianshui	34.08 (1)	1650	2016
GS-13	G22-14	Huating, Plingliang	6.43 (3)	1650	2016
GS-14	CYR34	Qinzhou, Tianshui	33.70 (1)	1500	2017
GS-15	CYR34	Jishishan, Linxia	23.40 (1)	2000	2019

^a^ CYR = Chinese Yellow Rust. Su11 and G22 series indicate race group with virulence to wheat genotypes Suwon 11 and Guinong 22 used as differential hosts, respectively.

**Table 2 plants-13-03518-t002:** Disease severity of *Puccinia striiformis* f. sp. *tritici* isolates on infected leaves of susceptible wheat cv. “Mingxian 169” seedlings after 16 days of incubation at 16 °C and 22 °C.

Code	Isolate	Year	Disease Severity ^a^	
			16 °C	22 °C
GS-1	Su11-4	2005	36.7abc	12.2ab
GS-2	Su11-7	2005	38.3abc	10.2a
GS-3	Su11-5	2005	30.3a	9.5a
GS-4	CYR33	2005	42.3abc	29.0cd
GS-5	CYR32	2005	38.7abc	27.7cd
GS-6	Su11-4	2010	41.7abc	13.0ab
GS-7	Su11-35	2010	45.7c	13.2ab
GS-8	CYR33	2010	44.3bc	23.0bcd
GS-9	CYR32	2010	43.3abc	29.3cd
GS-10	CYR33	2016	36.3abc	26.8cd
GS-11	CYR32	2016	45.0c	31.8d
GS-12	CYR34	2016	48.5c	35.0d
GS-13	G22-14	2016	47.8c	24.1bcd
GS-14	CYR34	2017	30.7ab	17.7abc
GS-15	CYR34	2019	39.5abc	28.6cd
Average			40.6	22.1

^a^ Disease severity was evaluated by the method described by Kiani et al. [26]. The data of disease severity of each isolate are average value of three repetitions. Significant differences are indicated with lowercase letters (*p* < 0.05).

**Table 3 plants-13-03518-t003:** Analyses of variance of disease severity of *Puccinia striiformis* f. sp. *tritici* isolates collected in 2005, 2010, and post-2016 on infected leaves of susceptible wheat cv. “Mingxian 169” seedlings with urediospores for 16 days of incubation at different temperatures.

Subgroup of Isolates in Collection Year	Disease Severity ^a^	
	16 °C	22 °C
2005	37.3b	17.7b
2010	42.3a	21.1b
Post-2016	42.3a	27.5a

^a^ Data of disease severity indicate the average value of three repetitive treatments. Different lowercase letters indicate that there was a significant difference among years based on least significant difference (LSD) test at the level of 0.05 (*p* < 0.05).

**Table 4 plants-13-03518-t004:** Analyses of variance of disease severity of *Puccinia striiformis* f. sp. *tritici* isolates on infected leaves of susceptible wheat cv. “Mingxian 169” seedlings with urediospores at 16 days of inoculation.

Effects and Interactions	DF ^a^	F-Value	*p* ^b^
Temperature	1	179.6	***
Overall Isolate	14	3.1	***
Temperature × Year	4	3.0	*
Temperature × Isolates	14	3.9	***
Isolates at 16 °C	14	3.9	***
Isolates at 22 °C	14	11.5	***

^a^ DF = Degrees of freedom. ^b^ A single asterisk (*) indicated that there was a significant difference (*p* < 0.05). Three asterisks (***) indicate that there was an extremely significant difference (*p* < 0.001).

**Table 5 plants-13-03518-t005:** Analyses of variance of germination percentage of urediospores of *Puccinia striiformis* f. sp. *tritici* isolates for 4, 8, 12, and 24 h of incubation at 16 °C.

Code	Year	Isolate	Germination Percentage of Urediospores (%)			
			4 h	8 h	12 h	24 h
GS-1	2005	Su11-4	0.0a	0.1a	0.2a	0.7a
GS-2	2005	Su11-7	2.9ab	5.6b	4.7ab	7.2ab
GS-3	2005	Su11-5	0.0a	0.0a	0.3a	0.1a
GS-4	2005	CYR33	10.2cd	12.3c	19.1cd	32.9def
GS-5	2005	CYR32	9.7cd	18.7d	9.7b	49.0gh
GS-6	2010	Su11-4	0.4a	0.7ab	1.9a	4.3ab
GS-7	2010	Su11-35	2.2ab	5.1ab	3.9ab	9.5ab
GS-8	2010	CYR33	6.6abc	15.5cd	24.4de	43.9fg
GS-9	2010	CYR32	7.2bcd	16.4cd	27.1ef	38.4efg
GS-10	2016	CYR33	5.5abc	12.1c	17.8c	24.6cd
GS-11	2016	CYR32	10.5cd	13.3c	19.7cd	25.9cde
GS-12	2016	CYR34	13.7d	17.1cd	31.7f	59.6h
GS-13	2016	G22-14	7.3bcd	14.7cd	28.7ef	43.4fg
GS-14	2017	CYR34	0.0a	0.3ab	1.9a	46.2g
GS-15	2019	CYR34	0.0a	0.0a	0.0a	15.3bc
Average			5.1	8.8	12.7	26.7

Different lowercase letters indicate a significant different between temperature and host (wheat cv. “Mingxian 169”) at a level of 0.05 (*p* ≤ 0.05).

**Table 6 plants-13-03518-t006:** Analyses of variance of germination percentage of urediospores of *Puccinia striiformis* f. sp. *tritici* isolates collected in 2005, 2010, and Post-2016 at 16 °C.

Subgroup of Isolates in Collection Year	Germination Percentage of Urediospores (%)
2005	20.7b
2010	21.3b
Post-2016	38.0a

Different lowercase letters indicate a significant difference among years based on least significant difference (LSD) test at a level of 0.05 (*p* < 0.05).

**Table 7 plants-13-03518-t007:** Latent period of *Puccinia striiformis* f. sp. *tritici* isolates on infected leaves of wheat cv. “Mingxian 169” seedlings with urediospores for 16 days of inoculation at 18 °C and 22 °C.

Code	Isolate	Year	Latent Period at Different Temperatures (d) ^a^	
			18 °C	22 °C
GS-1	Su11-4	2005	10.8abc	11.5bc
GS-2	Su11-7	2005	11.3c	12.2c
GS-3	Su11-5	2005	10.8abc	11.5bc
GS-4	CYR33	2005	10.2abc	10.8ab
GS-5	CYR32	2005	10.5abc	10.9ab
GS-6	Su11-4	2010	10.8abc	11.4bc
GS-7	Su11-35	2010	10.9bc	11.6bc
GS-8	CYR33	2010	10.2abc	10.7ab
GS-9	CYR32	2010	9.7a	10.2a
GS-10	CYR33	2016	10.4abc	10.8ab
GS-11	CYR32	2016	10.1ab	10.7ab
GS-12	CYR34	2016	11.1bc	11.8bc
GS-13	G22-14	2016	10.5abc	10.8ab
GS-14	CYR34	2017	10.0ab	10.8ab
GS-15	CYR34	2019	10.0ab	10.8ab
Average			10.5	11.1

^a^ Different lowercase letters indicate a significant difference among temperature and host interactions (wheat cv. “Mingxian 169”) at a level of 0.05 (*p* ≤ 0.05). Data of latent period are the average value of three repetitions.

**Table 8 plants-13-03518-t008:** Analyses of variance of latent period of *Puccinia striiformis* f. sp. *tritici* isolates collected in 2005, 2010, and post-2016 on infected leaves of susceptible wheat cv. “Mingxian 169” seedlings with urediospores incubation time at different temperatures.

Subgroup of Isolates in Collection Year	Latent Period at Different Temperatures ^a^	
	18 °C	22 °C
2005	10.38a	11.00a
2010	10.65a	11.28a
Post-2016	10.37a	11.00a

^a^ Different lowercase letters indicate a significant difference among years based on least significant difference (LSD) test at a level of 0.05 (*p* < 0.05).

## Data Availability

The original contributions presented in this study are included in the article. Further inquiries can be directed to the corresponding authors.

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
