# Peer review of "Aggressiveness of Puccinia striiformis f. sp. tritici Isolates at High Temperatures: A Study Case in Core Oversummering Area of Gansu as Inoculum Source"

_plants, 2024, doi:10.3390/plants13243518_

Round 1

Reviewer 1 Report (Previous Reviewer 2)

Comments and Suggestions for Authors

Dear Author 

The author uses appropriate scale to measure disease severity in the seedling stage. The severity scale is used for stripe rust seedling stage dependent on the pustule sporulation does not based on lesions. Stripe rust symptoms do not formulate lesions. 

The other thing is the type of sporulation of the infected area is important. With high temperature infection will still perform but the sporulation will be reduced. Thant mean the rust infection will not propagate in the next cycle. But if the stripe races that the author find is tolerate the high temperature it should be sporulate as in low temperature or similar to that.

Infection type will give measurement on how density the sporulation is on the infected area, it's very critical in this type of study. Please check the paper below.

For this reason i will reject the manuscript.

Thanks

Chen, X. Pathogens which threaten food security: Puccinia striiformis, the wheat stripe rust pathogen. Food Sec. 12, 239–251 (2020). https://doi.org/10.1007/s12571-020-01016-z

Author Response

The author uses appropriate scale to measure disease severity in the seedling stage. The severity scale is used for stripe rust seedling stage dependent on the pustule sporulation does not based on lesions. Stripe rust symptoms do not formulate lesions. The other thing is the type of sporulation of the infected area is important. With high temperature infection will still perform but the sporulation will be reduced. Thant mean the rust infection will not propagate in the next cycle. But if the stripe races that the author find is tolerate the high temperature it should be sporulate as in low temperature or similar to that.

Infection type will give measurement on how density the sporulation is on the infected area, it's very critical in this type of study. Please check the paper below.

Response: Thank you for your critical comments and suggestions. We revised the scale for measuring disease severity at seedling stage according to the ratings method described in the literature (Kiani et al. 2021. control of stripe rust of wheat using indigenous endophytic bacteria at seedling and adult plant stage. Sci. Rep., 11:14473), and the literature was added in reference list. We re-ordered the reference list and revise origina order in the text. Additionally, wheat cv. Mingxian 169, a highly susceptible to all known races of the stripe rust in China, any races of the rust infect plants o the wheat cv. do not produce chlorosis and necrosis. Thus, disease severity, viz. rust severity, is routinely used to measure sporulation area, and overall races used produce high infection types on wheat cv. Mingxian 169, without difference in infection type. Although races are tolerant to high temperatures, it can sporulate normally and had a little stronger ability for sporulation than non-high temperature-tolerant races in addition to differences in latent period, and high spore germination in high temperatures.

Sporulation intensity is an index for evaluating sporulation ability. This index is used for assessing sensitivity and resistance to fungicide (Zhan et al., 2022. Sensitivity and resistance risk assessment of Puccinia striiformis f. sp. tritici to Triadimefon in China, Plant Disease, 106:1690-1699). No difference was observed among fungicide-sensitive and -resistant isolates of the stripe rust in Zhan’s study. In our study, there was also no different among isolates races. We think it is unnecessary for adding this information. Additionally, according to previous studies on isolates to high temperatures (Milus et al., 2006, 2009; Loladze et al., 2014), this index did not used to evaluate for it in the present study. 

It is fact that low temperatures are favor for the development of the stripe rust. The rust can germinate at a range of 0.5-23 ℃, and optimal temperature for germination is 9-13℃. After penetration into host tissue, the rust can develop under average 23℃, and thus, even if the temperatures had a wide range such as 10-36℃ (average 23℃), which are used for evaluating the level of resistance of wheat cultivars under high temperatures, the rust can develop and produce urediospores. High temperatures may influence the quantity of uredia and urediospores, but not cease sporulation. We are persisting in working for stripe rust research for decades, according to our work experiences, no matter the urediospores being collected from high temperature in laboratory conditions, or from wheat fields in warmer weathers in late spring or early summer in the crop growing stage. The collected urediospore can germinate and infect wheat, we have not observed obvious difference with urediospores that we collected in the laboratory conditions.

The stripe rust is capable of extremely rapid reproduction. A single urediospore can produce approximately 2 million fresh urediospores in an asexual reproduction cycle, 8-10 days under favorable conditions. As long as urediospores can produce and urediospores can germinate and complete asexual cycle by infection and reinfection. Except that urediospores are directly exposed for a long time in high temperatures, and lose their lives.

Reviewer 2 Report (New Reviewer)

Comments and Suggestions for Authors

The manuscript should be able to improve more with the content, attracting more readers. If the authors can include at least one figure to present data, the manuscript will be more interesting.

Author Response

comments1:The manuscript should be able to improve more with the content, attracting more readers. If the authors can include at least one figure to present data, the manuscript will be more interesting.

Response: Thank you for your valuable suggestions. We made three figures for easily catch in addition to tables.

Reviewer 3 Report (New Reviewer)

Comments and Suggestions for Authors

Dear Authors,

The peer-reviewed manuscript entitled "Aggressiveness of Puccinia striiformis f. sp. tritici isolates to high temperatures: a study case in core oversummering area of ​​Gansu as inoclum source" raises an important problem related to the threat caused by fungal diseases caused by Puccinia striiformis f. sp. tritici on wheat plants. The authors conducted interesting and useful research from an agronomic point of view. Below are my suggestions for the manuscript:

- Introduction - I suggest adding a short information about wheat, because it is a strategic crop and there is no connection here with the described fungal disease

- The authors inappropriately cite studies by other authors in some places in the manuscript, e.g. Milus et al. (2006) - instead of the year of publication in brackets there should be a number (it should be corrected throughout the manuscript)

- Materials and Methods:

Subsection 4.1. - provide the manufacturer and parameters of the growing chamber

Subsection 4.2. - provide the developmental stage of wheat seedlings at the time they were inoculated (according to the selected scale)

- the manuscript should contain a separate Conclusions chapter containing the most important research results and application of the research results

Author Response

Comments1: The peer-reviewed manuscript entitled "Aggressiveness of Puccinia striiformis f. sp. tritici isolates to high temperatures: a study case in core oversummering area of Gansu as inoclum source" raises an important problem related to the threat caused by fungal diseases caused by Puccinia striiformis f. sp. tritici on wheat plants. The authors conducted interesting and useful research from an agronomic point of view. Below are my suggestions for the manuscript:

- Introduction - I suggest adding a short information about wheat, because it is a strategic crop and there is no connection here with the described fungal disease

Response: Thank you for the reviewer’s suggestions. We added this information. 

- The authors inappropriately cite studies by other authors in some places in the manuscript, e.g. Milus et al. (2006) - instead of the year of publication in brackets there should be a number (it should be corrected throughout the manuscript)

Response: Thank you for your carefulness. We changed and looked through the text of the manuscript for ensuring citations correct in format.

- Materials and Methods:

Subsection 4.1. - provide the manufacturer and parameters of the growing chamber

Response: Thank you for your suggestions. We added.

Subsection 4.2. - provide the developmental stage of wheat seedlings at the time they were inoculated (according to the selected scale)

Response: Thank you for your suggestions. The information on 10-day-old wheat seedlings is already present in this section. We re-organized the sentences.

- the manuscript should contain a separate Conclusions chapter containing the most important research results and application of the research results

Response:

Response: Thank you for your carefulness. We added the conclusion section in the text of the end of the discussion section.

Round 2

Reviewer 1 Report (Previous Reviewer 2)

Comments and Suggestions for Authors

Dear Author

The manuscript is improve and the author address all my comments. I will accepted as present.

Thanks

This manuscript is a resubmission of an earlier submission. The following is a list of the peer review reports and author responses from that submission.

Round 1

Reviewer 1 Report

Comments and Suggestions for Authors

Please find my comments in the attached MS.

Comments on the Quality of English Language

My comments on the English language can be found in the attached MS.

Author Response

Comments 1- The phrase "As a result" could be replaced with "The results indicate" for better clarity and flow. This change would enhance the scientific tone of the manuscript.

Response: Thank you for your suggestions. We made revision in abstract.

Comments 2- noticed a bit of redundancy in the second paragraph with the phrase "wheat stripe rust is a destructive fungal disease in wheat." Since "wheat stripe rust" already implies it affects wheat (from the first paragraph), you might consider revising it to eliminate this repetition for clarity.

Response: Thank you for your valuable comments. We revised these sentences. We moved the sentence to first paragraph and added several sentences in this paragraph.

Comments 3- Is this your hypothesis? If not please provide reference for it.

Response: Thank you for your critical comments. This is the fact but not a hypothesis. We added references cited in the text.

Comments 4- I noticed that the Introduction shifts from specific Chinese races ( in the above paragraph) to broader global patterns in stripe rust. This transition moves from a detailed focus on local data to a more general overview, which may disrupt the flow of the narrative. It might be beneficial to clearly indicate this shift by adding a transitional phrase or sentence that connects the significance of the Chinese races to their implications or relevance on a global scale. This would enhance coherence and guide the reader through the argument more smoothly.

Response: Thank you for your valuable and critical comments. We re-organized this paragraph for easy to understand widespread of the rust for accelerating stripe rust epidemics worldwide.

Comments 5- The phrase "we supposed that" to be somewhat informal for this context. It may be more effective to use "hypothesized" or "postulated," as these terms convey a stronger scientific basis for the assumption being made. Additionally, the sentence could benefit from clearer phrasing to enhance clarity. For example, consider rephrasing to: "Therefore, we hypothesized that the incidences of the disease in China could be attributed to Pst isolates that have adapted to high temperatures, similar to those reported in the United States and elsewhere, demonstrating increased aggressiveness under warm conditions.

Response: Thank you for your suggestions and comments for the revision. We made revised this section for easy to understand.

Comments 6-Please mention how old these seedlings were.

Response: we made revision.

Comments 7 -The sentence could be improved for clarity and readability... Sixteen days after inoculation with urediniospores and incubation at 16°C.

Response: Thank you for your suggestions. We changed.

Comments 8 -The data presented in Table 5 is comprehensive and provides valuable insights into the germination percentage of urediospores (GPU) across different time points and isolates. However, I would suggest streamlining the text to avoid redundancy and to focus on highlighting the key findings, such as trends or significant differences between isolates. This would make the narrative more concise and engaging for the reader, while still encouraging them to refer to the table for specific details.

Response: Thank you for your critical comments. We re-organized the text for easy to catch key findings according to Table 5.

Comments 9 -Why Table 4 was mentioned after Table 5? Also please refer to my comment above.

Response:Thank you for your suggestions. We made change in the text.

-In my opinion, tables 7, and 10 may not be necessary as they do not significantly contribute to the overall conclusions of the study. The trends presented in these tables could be summarized more concisely in the text. Removing these tables may help streamline the manuscript and improve its readability without losing essential information.

Response: Thank you for your valuable comments and good suggestions. We deleted tables 7 and 10.

Comments 10-Why these are in a different font through out the MS?

Response: Thank you for your careful reviewing. We made it correct in format.

Comments 11-This was repeated several times and mentioned in the introduction as well. Please focus on interpreting the results.

Response: Thank you for your critical comments. We deleted it and re-organized this paragraph.

Comments 12-This does not belong in Discussion. Please mention any future work that needs to be done or any drawbacks of this study here.

Response: Thank you for your critical and valuable comments. We re-organized this section.

Comments 13-I was wondering if you could clarify how the Pst races were determined and differentiated, particularly between the 'old' and 'new' races. It would be helpful to know the specific criteria or methods used to classify the isolates into these groups. This information would enhance the understanding of your comparative study.

Response: Thank you for your critical comments. We added more detailed information for consideration.

Reviewer 2 Report

Comments and Suggestions for Authors

Dear Author

Please find my comments on the general points in this study:

-The Author does not follow the right scoring protocol for rating stripe rust in the seedling stage. The protocols the author follows in this study are for the adult stage and it's completely inappropriate for the seedling stage.

-The study should continue to the adult stage to see how the disease developed under different temperature conditions. The seedling study will not approve the author's point.

-The difference in the germination rate of stripe rust spores may related to either non-proper storage or sometimes because it old culture. Autore needs to first standardize the rate of germination under standard protocol for all races used in this study and after that look at the effect of different temp on the rate of germination and disease development. 

Thanks

Author Response

Comments 1-The Author does not follow the right scoring protocol for rating stripe rust in the seedling stage. The protocols the author follows in this study are for the adult stage and it's completely inappropriate for the seedling stage.

Response: Thank you for your critical comment. We highly agreed with the comments from the reviewer. Due to our carelessness, this literature cited was used inappropriately.  We made correction about the rating method for stripe rust. We measured lesion area by observing the percentage of the infected area accounting for the total leaf area. So, data we scored in this work were not affected. The literature cited we used in this section is deleted.

Comments 2-The study should continue to the adult stage to see how the disease developed under different temperature conditions. The seedling study will not approve the author's point.

Response: Thank you for your valuable suggestions. Before we started to study this work, we looked through a previous literature that first reported on aggressiveness of the rust in the USA (Milus et al., 2006). In this paper, highly susceptible wheat cultivars were used to evaluate aggressiveness of the races at seedling stage. So, we adopted the method by this paper and measure data at seedling stage. In addition, wheat cv. Mingxian 169 we used in this work is highly susceptible for all known races of the rust in China and routinely used to recover stripe rust samples and multiple spores. Therefore, there is no difference for infection at either seedling stage or adult plant stage.

Comments 3-The difference in the germination rate of stripe rust spores may related to either non-proper storage or sometimes because it old culture. Author needs to first standardize the rate of germination under standard protocol for all races used in this study and after that look at the effect of different temp on the rate of germination and disease development.

Response: Thank you for your comments. In this study, overall isolate, regardless of new or old isolates, each was first recovered on highly susceptible wheat cv. Mingxian 169 and multiplied spores by re-inoculation using routine methods. Abundant fresh spores were collected and then used to measure spore germination. Therefore, there are difference of overall isolates in spore germination rate, which is reasonable due to their biological nature to some environmental factors influencing the development, i.e. temperatures.
